

# Response of soil arbuscular mycorrhizal fungal community to chemical fertilization in sugarcane

Yi-Hao Kang[1,*], Shang-Tao Jiang[2,*], Qian Wang[1], Ying-Jie Nong[1,3], Juan Song[1], Dong-Ping Li[1], Yun-Ying Wen[1], Jie Xu[1], Ting-Su Chen[1], Jin-Lian Zhang[1] and Yang-Rui Li[4]

[1] Microbiology Research Institute, Guangxi Academy of Agricultural Sciences, Nanning, Guangxi, China
[2] Jiangsu Key Laboratory for Bioresources of Saline Soils, Jiangsu Synthetic Innovation Center for Coastal Bio-agriculture, School of Wetlands, Yancheng Teachers University, Yancheng, Jiangsu, China
[3] Guangxi State Farms Yongxin Animal Husbandry Group Company, Nanning, Guangxi, China
[4] Key Laboratory of Sugarcane Biotechnology and Genetic Improvement (Guangxi), Ministry of Agriculture and Rural Affairs, Guangxi Key Laboratory of Sugarcane Genetic Improvement, Sugarcane Research Institute, Guangxi Academy of Agricultural Sciences, Nanning, Guangxi, China
* These authors contributed equally to this work.

Corresponding authors
Jin-Lian Zhang,
zhangjinlian1@126.com
Yang-Rui Li, liyr@gxaas.net

## ABSTRACT

Most sugarcane cultivation areas in China have undergone over 30 years of continuous monocropping, and long-term chemical fertilizer application has led to severe soil degradation. Arbuscular mycorrhizal fungi (AMF) play a crucial role in promoting plant nutrient uptake, enhancing plant stress tolerance, and improving soil quality and restoration. However, in agroecosystems, AMF are susceptible to the effects of cultivation, fertilization, and other factors. The goal of this study was to investigate the influence of chemical fertilization on the AMF community in the rhizosphere soil of plant crop of sugarcane. In this study, sugarcane varieties GT58 and GT29 were selected for experiment. Four different chemical fertilization rates were established using controlled-release compound chemical fertilizer (N: P: K = 17:7:17): T1 (0.0 kg/ha), T2 (562.5 kg/ha), T3 (1,125.0 kg/ha), and T4 (2,250.0 kg/ha). The results showed that different fertilization rates significantly affected the cane yield in GT58. T3 and T4 increased the cane yield by 12.67% and 11.11%, respectively, compared to the control T1 ($P < 0.05$). The impact on the cane yield in GT29 was not significant. The diversity indices of root-associated AMF in GT58 (Chao index and Shannon index) varied significantly in different fertilization rates. T3 had the highest diversity, showing no significant difference from T1 and T2 but significantly higher than T4. For GT29, there were no significant differences in the diversity indices of root-associated AMF among different fertilization rates. Analyzing the differential species in root-associated soil with different fertilization rates in GT58 at the OTU level revealed that T3 had significantly higher abundances for 5 OTUs compared to the other treatments, and all the differentially abundant species belonged to *Glomus*. The Mantel analysis revealed that the genus *Acaulospora* was significantly and positively correlated with millable stalks ($P < 0.001$), and significantly and positively correlated with tiller number ($P < 0.05$). The genus *Scutellospora* showed a significant positive correlation with sprouting rate ($P < 0.05$).

However, the other AMF genera did not show significant correlations with the agronomic traits of sugarcane. In summary, different chemical fertilization rates significantly affected the cane yield in GT58 and the AMF community in the rhizosphere soil, but not in GT29, which suggests that sugarcane chemical fertilization should consider different sugarcane varieties and the diversity of AMF communities in soil.

## INTRODUCTION

Sugarcane, a highly efficient $C_4$ plant, is one of the most significant global sugar crops in tropical and subtropical regions (*Li et al., 2006*). Currently, sugarcane provides around 80% of the world's sugar and 35% of ethanol production. It serves as a crucial pillar industry for economic development and improving the quality of life in developing countries (*Food and Agriculture Organization of the United Nations, 2020*). China ranks as the world's third-largest producer of sugarcane (*Tayyab et al., 2021*). The major sugarcane cultivation regions in China lie in the tropical and subtropical areas, that is, south of the 25°N latitude, mainly in Guangxi, Yunnan, Guangdong and Hainan provinces. Guangxi is the biggest sugarcane cultivation area in China. In 2020, the sugarcane planting area in Guangxi reached an impressive 875,000 ha, accounting for 64.7% of China's total sugarcane planting area (*China National Bureau of Statistics, 2021*). The sugarcane cultivation in China confronts issues such as excessive chemical fertilizer application, nitrogen bias, low chemical fertilizer utilization efficiency, *etc.* (*Lin, 2017*). These problems not only lead to severe soil compaction and acidification and deceased cane yields, but also harm the microenvironment around sugarcane roots (*Yi et al., 2014*).

Arbuscular mycorrhizal fungi (AMF) are widely in the soil environment, capable of forming symbiotic relationships with the roots of over 80% of higher plants (*Smith & Read, 2008*). AMF can enhance the resistance of host plants to stress (*Chen et al., 2009*; *Rasmussen et al., 2011*; *Lang et al., 2022*; *Schröder, Mohri & Kiehl, 2019*). They also play a role in improving soil aggregate structure and soil restoration (*Zhang et al., 2022*). Sugarcane is a mycorrhizal-dependent plant, and it forms endomycorrhizal symbiosis with AMF (*Sivakumar, 2013*; *Reis, de Paula & Döbereiner, 1999*; *Nasim et al., 2008*; *Rokni, Goltapeh & Alizadeh, 2010*). The rhizosphere soil of sugarcane in Guangxi has rich resources of AMF. A total of Seven families, twelve genera, and twenty-four species of AMF were identified in the rhizosphere soil of sugarcane (*Chen, Zhang & Long, 2015*). AMF can promote sugarcane growth, and sugarcane inoculated with AMF showed better performance in plant height, leaf number, stem diameter, and root development (*Wang et al., 1995*; *Chen et al., 2011*). Field experiments have demonstrated AMF increase soil pH and enhance nutrient absorption by sugarcane roots, leading to cane yield increases by 10% to 24% and reducing chemical fertilizer application by over 30% (*Chen et al., 2013*; *Wu et al., 2015*).

In agricultural ecosystems, AMF are susceptible to the cultivation and chemical fertilization. Our previous study (Nong et al., 2023) showed the mixing planting of two sugarcane varieties resulted in a decrease in the relative abundance of *Glomus* but an increase in the relative abundance of *Acaulospora*. Based on this, the present study was further carried out to investigate the growth of sugarcane and the community of AMF in the rhizosphere soil under different chemical fertilizations for the goal to understand the rhizosphere AMF community structure of different sugarcane varieties with varied ratoon ability and to provide a reference for exploring agronomic measures to improve the cane yield in sugarcane production and to reduce the production cost, which would be good for sustainable development of the sugarcane industry.

## MATERIALS AND METHODS

### Study site and experimental design

The experiment was conducted in 2020 at the experimental field of the Li-Jian Scientific Research Station of Guangxi Academy of Agricultural Sciences (23°14′N, 108°33′E), where average annual temperature is 21.7 °C, average annual humidity is 78.6%, average annual precipitation is 85.3 mm, and average annual sunshine length is 118.0 h. The field experiment was designed with randomized block with six duplicates. Each experimental plot consisted of five rows, with a row length of 7 m, a row spacing of 1.2 m and planting depth of 30 cm. Three rows of the same sugarcane variety were planted between different plots to serve as isolation areas. The basic physicochemical properties of the soil used were as follows: pH 5.1, organic matter 32.3 g/kg, total nitrogen 2.3 g/kg, total phosphorus 1.8 g/kg, total potassium 10.9 g/kg, water soluble nitrogen 51.97 mg/kg, available phosphorus 73.2 mg/kg, and available potassium 110 mg/kg. Two sugarcane varieties were chosen: Gui-Tang 58 (GT58), known for its extensive cultivation, and Gui-Tang 29 (GT29), characterized by strong ratoon and tillering ability. The specific characteristics of the sugarcane varieties are as follows:

GT58: This variety was selected from the cross of YT85-177 × CP81-1254. It is currently a widely cultivated variety. Its distinctive characteristics include medium plant height, large stalk diameter, more millable canes, slightly purplish-red leaf sheaths, underdeveloped hair groups, and easy leaf shedding. It has strong ratoon ability, strong adaptability, upright stalk, and lodging resistance, and lacks aerial roots. GT58 is known for high and stable cane yields, medium maturity, high sugar content, high resistance to smut disease, but average drought resistance.

GT29: The parentage of this variety is YC94-46 × ROC22. It has strong ratoon ability and can have more than nine ratoons in production. Its characteristics include erect and compact plant structure, easily detached leaves, intermediate stalk, uniform canes, early ripening, and high sugar. It has excellent sprouting ability and strong tillering ability, strong cold tolerance, and is highly resistant to smut.

The sugarcane planting date was January 8, 2020. Due to variations in tillering in different sugarcane varieties, especially the strong tillering ability of GT29, 120,000 buds/ha for GT58 and 75,000 buds/ha for GT29 were used, respectively. Chemical fertilizer was applied for two times. At planting, 40% of the base chemical fertilizer was applied, and 60%

of the additional chemical fertilizer was applied just before earthing-up at early elongation stage. Considering the common chemical fertilization about 1,500 kg/ha used in commercial sugarcane production in Guangxi and referring to recent field research results related to sugarcane mycorrhizal studies, the experiment employed four different chemical fertilization treatments: T1 (0.0 kg/ha), T2 (562.5 kg/ha), T3 (1,125.0 kg/ha), and T4 (2,250.0 kg/ha). The chemical fertilizer used was controlled-release compound chemical fertilizer with an N: P: K ratio of 17:7:17 (product of Hebei Tian-Ren Chemical Industry Co., Ltd., Qinhuangdao, China). The crop was managed the same as for common commercial production, and harvest was done in December 2020.

## Sugarcane agronomic trait measurements

After emergence, the sprouting rate of sugarcane was investigated on April 10, 2020. On May 26, 2020, the number of tillers per plant was surveyed. On November 20, 2020, the number of millable stalks, that is, the plant height was 1.3 m or more, was investigated. In December 2020, the crop was harvested, and 20 plants with similar growth were selected from the middle three rows. Plant height was measured using a tape measure, stalk diameter was measured using a vernier caliper, and hardness was measured using a hardness tester (MYT-1). After harvested, all the canes within the same plot were collected and weighed, and the data were used for estimation of the cane yield per hectare.

## Soil and root sample collection

On December 10, 2020, samples of sugarcane roots and rhizosphere soil from 0–30 cm soil layer were collected using a five-point sampling method, that is, the five points of the middle and four corners in a plot. The soil around the roots of five sugarcane plants were collected each plot and thoroughly mixed as a soil sample. There were five replicates.

Soil sample collection: At each sampling point, 0.5–1.0 kg of rhizosphere soil from 0–30 cm depth was collected and thoroughly mixed. Using the quartering method, 0.5–1.0 kg of soil was taken from each plot as the soil sample. The soil samples were sieved through a 20–mesh sieve, and 800–1,000 mg portions were put into 2 mL centrifuge tubes. Each soil sample was divided into three tubes and stored at −80 °C for subsequent total DNA extraction.

Root sample collection: Concurrently with soil sampling, root samples were collected. At each sampling point, after rough separation of the collected roots and rhizosphere soil, the roots were mixed as one sample that contained roots of varied sizes. The root sample treatment was the same as described in the previous report (*Zhang et al., 2024*).

## Root AMF colonization assessment

Root staining was conducted to observe AMF colonization structures. The root staining method was referred to *Liao et al. (2016)*. The cleaned roots were cut into about 1 cm long segments, placed in embedding boxes, and placed in 20% KOH solution for 90–120 min in a 90 °C water bath. The root AMF colonization was determined according to *Nong et al. (2023)* using the Nikon microscope Eclipse Ci-L and DS-Ri2 to observe the root

colonization, and the MYCOCALC software to calculate the colonization rate based on the results.

## Soil AMF high-throughput sequencing

FastDNA® Spin Kit for soil DNA extraction was used to extract DNA from soil samples, and the specific operation was conducted according to the instructions. Nested PCR was performed to amplify the extracted AMF DNA. Small subunit rRNA (SSU) was selected for amplification. The high-throughput sequencing followed the procedure described in our previous report (*Nong et al., 2023*).

## Data analyses

Trimmomatic 0.38 was used for quality control of raw sequencing data, followed by paired-end assembly using Flash 1.2.11. UPARSE 11 was employed to cluster sequences into distinct Operational Taxonomic Units (OTUs) based on a 97% similarity threshold. The resulted OTUs were aligned to the MaarjAM database for sequence matching and annotated using RDP Classifier with a confidence threshold of 0.7. Unclassified sequences at the phylum level were filtered out. The OTU table was normalized to minimum sample sequence counts, and subsequent analyses were performed. Mothur was used for α-diversity analysis, displaying species diversity with the Shannon index and richness with the Chao1 index. The Bray-Curtis dissimilarity matrix was built to estimate the β-diversity distances, Principal Coordinates Analysis (PCoA) was done to depict the influence of diverse fertilization treatments on the AMF community around sugarcane roots.
The Adonis test was done to assess the significance in AMF communities in different fertilization treatments. The agronomic traits were compared using one-way ANOVA and Duncan's test. Correlations between sugarcane agronomic traits and diverse AMF genera were examined through Mantel tests, while Pearson correlation analysis was done to explore the relationships between various sugarcane agronomic traits.

# RESULTS AND ANALYSIS

## Effects of chemical fertilization on agronomic traits of sugarcane

The agronomic traits in sugarcane were influenced by chemical fertilization. For GT58, the sprouting rate was significantly affected by different chemical fertilization rates. T4 showed no significant difference from T2 ($P = 0.067 > 0.05$) but exhibited significant increases than T1 and T3 ($P = 0.012 < 0.05$). The tiller number was significantly impacted by different fertilization treatments. The number of tillers in T1, T2, T3 and T4 were 138.23, 146.40, 152.28 and 153.83, respectively. No significant differences were observed between T4, T3, and T2, however, both T4 and T3 showed significant increases than T1 by 11% and 10%, respectively. Moreover, the different fertilization rates significantly influenced the millable stalks in both GT58 and GT29. For GT58, there was no significant difference between T4 and T3, yet that in T4 was significantly higher than T1 and T2 by 13% and 8%, respectively. For GT29, there were no significant differences in T4, T3, and T2, however, T4 and T3 both showed a significant increase than T1 by 6%. Conversely, the plant height and stalk

**Table 1 The agronomic traits of two sugarcane varieties under different chemical fertilization rates.**

| | Treatment | Sprouting rate (%) | Tillers (%) | Millable stalks/plot | Plant height (cm) | Stalk diameter (mm) |
|---|---|---|---|---|---|---|
| GT58 | T1 | 56.03 ± 4.28b | 138.23 ± 10.15b | 69.96 ± 2.54c | 329.21 ± 24.32a | 27.69 ± 0.84a |
| | T2 | 58.37 ± 3.61ab | 146.4 ± 10.22ab | 73.17 ± 3.3bc | 337.67 ± 14.57a | 27.5 ± 0.91a |
| | T3 | 57.23 ± 4.93b | 152.28 ± 6.9a | 76.71 ± 3.52ab | 344.68 ± 14.34a | 27.13 ± 1.2a |
| | T4 | 62.87 ± 4.57a | 153.83 ± 11.27a | 79.21 ± 2.73a | 339.26 ± 15.28a | 26.56 ± 1.22a |
| GT29 | T1 | 53.07 ± 6.45a | 166.28 ± 20.48a | 86.83 ± 3.92b | 332.86 ± 15.74a | 25.6 ± 0.54a |
| | T2 | 49.4 ± 5.82a | 178.02 ± 12.46a | 88.88 ± 2.56ab | 339.97 ± 5.7a | 25.48 ± 0.66a |
| | T3 | 54.1 ± 3.61a | 181.4 ± 14.13a | 91.92 ± 4.31a | 327.45 ± 7.06a | 24.56 ± 0.43a |
| | T4 | 56.07 ± 4.05a | 172.45 ± 11.27a | 91.71 ± 3.52a | 328.91 ± 16.27a | 25.42 ± 1.29a |

Note:
Different lowercase letters indicate the significant differences at $P < 0.05$ in the same column. T1, T2, T3, and T4 represent the treatments with 0.0, 562.5, 1,125.0, and 2,250.0 kg/ha controlled-release compound chemical fertilizer with an N: P: K ratio of 17:7:17 (product of Hebei Tian-Ren Chemical Industry Co., Ltd., Qinhuangdao, China), respectively.

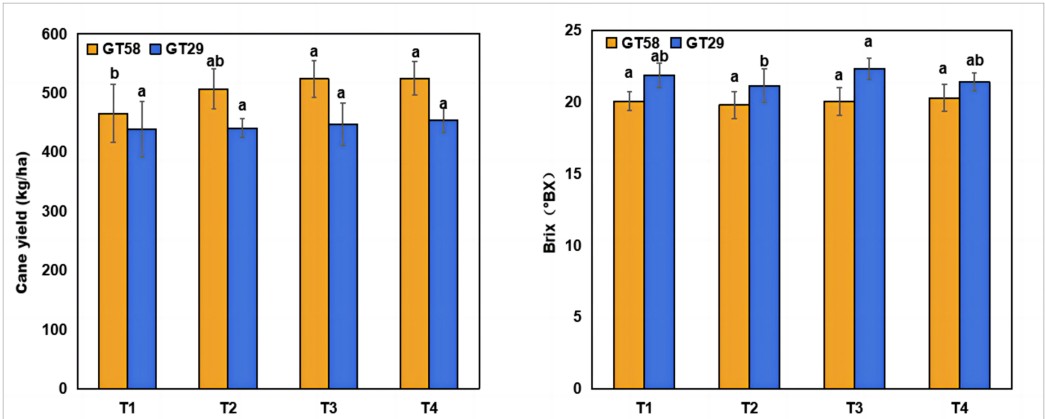

**Figure 1 Cane yield and brix in cane juice under different chemical fertilization rates.** Different lowercase letters above the columns indicate the significant differences at $P < 0.05$. T1, T2, T3, and T4 represent the treatments with 0.0, 562.5, 1,125.0, and 2,250.0 kg/ha controlled-release compound chemical fertilizer with an N: P: K ratio of 17:7:17 (product of Hebei Tian-Ren Chemical Industry Co., Ltd., Qinhuangdao, China), respectively.

diameter in both GT58 and GT29 were not significantly affected by different fertilization rates (Table 1).

The impact of different chemical fertilization rates on cane yield and brix is illustrated in Fig. 1. Fertilization significantly affected the cane yield in GT58, while did not in GT29. For GT58, there was no significant difference in cane yield between T4, T3, and T2, however, both T4 and T3 show significant higher cane yield by 13% than T1. Different fertilization rates significantly influenced the brix in GT29, but not in GT58. For GT29, T3 exhibited the highest brix (22.31°Bx), which was 6% higher than that in T2 (22.13°Bx).

## Mycorrhizal colonization in sugarcane roots with different chemical fertilization rates

It showed that different chemical fertilization treatments for both sugarcane varieties established favorable symbiotic relationships with AMF, with colonization rates exceeding 90%. However, there was no significant difference in colonization rates among different

**Table 2 Mycorrhizal colonization in sugarcane roots under different chemical fertilization rates.**

| Variety | Fertilization | Colonization rate (%) | Colonization intensity (%) | Relative colonization intensity (%) | Relative arbuscular abundance (%) | Arbuscular abundance (%) |
|---------|---------------|----------------------|----------------------------|-------------------------------------|-----------------------------------|--------------------------|
| GT58 | T1 | 95.00 ± 10.00a | 38.99 ± 8.65a | 39.26 ± 8.15a | 24.51 ± 6.21a | 10.91 ± 2.69a |
|      | T2 | 98.33 ± 4.08a | 35.92 ± 6.6ab | 35.92 ± 6.6a | 16.45 ± 1.45b | 6.78 ± 1.9ab |
|      | T3 | 95.00 ± 3.16a | 26.4 ± 6.72b | 28.03 ± 6.6a | 10.61 ± 3.94b | 4.59 ± 1.22b |
|      | T4 | 90.00 ± 12.25a | 27.65 ± 9.28ab | 32.76 ± 8.85a | 24.98 ± 4.51a | 8.91 ± 5.77ab |
| GT29 | T1 | 93.33 ± 6.83a | 24.28 ± 8.47ab | 26.18 ± 8.06ab | 25.37 ± 12.53ab | 8.62 ± 6.69ab |
|      | T2 | 95.83 ± 4.92a | 28.92 ± 6.82ab | 30.16 ± 6.77ab | 30.99 ± 9.53a | 11.69 ± 3.13ab |
|      | T3 | 96.67 ± 4.08a | 32.74 ± 7.01a | 33.55 ± 5.95a | 24 ± 6.67ab | 4.6 ± 1.73b |
|      | T4 | 94.00 ± 8.22a | 22.28 ± 5.15b | 21.74 ± 6.41b | 12.84 ± 9.79b | 4.42 ± 3.18b |

Note:
Different lowercase letters indicate the significant differences at $P < 0.05$, while different uppercase letters indicate the extremely significant differences at $P < 0.01$ in the same column. T1, T2, T3, and T4 represent the treatments with 0.0, 562.5, 1,125.0, and 2,250.0 kg/ha controlled-release compound chemical fertilizer with an N: P: K ratio of 17:7:17 (product of Hebei Tian-Ren Chemical Industry Co., Ltd., Qinhuangdao, China), respectively.

fertilization treatments ($P = 0.037 < 0.05$). For GT58, varying fertilization rates significantly influenced the colonization intensity, arbuscular abundance, and relative arbuscular abundance in sugarcane roots. Notably, T1 showed significant differences from T3 while no significant differences from T2 and T4. For GT29, colonization intensity and relative colonization intensity exhibited the pattern T3 > T2 > T1 > T4, showing a 47% increase in colonization intensity in T3 compared to that in T4. Arbuscular abundance and relative arbuscular abundance followed the pattern T2 > T1 > T3 > T4 in GT29, and T2 and T4 exhibited significant differences, demonstrating a 164% increase in arbuscular abundance and a 141% increase in relative arbuscular abundance in T2 compared to those in T4 (Table 2).

## AMF species composition under different chemical fertilization rates

A total of 258 OTUs were identified in the four different chemical fertilization levels in GT58. The number of OTUs for the fertilization level T1, T2, T3, and T4 was 70, 65, 68, and 55, respectively. Among them, there were three unique OTUs in T1, none in T2, 5 in T3, and none in T4. Additionally, there were 44 common OTUs shared for all the treatments, representing 17.05% of the total OTUs. For sugarcane variety GT29, a total of 273 OTUs were detected in the four chemical fertilization treatments. The OTU counts for T1, T2, T3, and T4 were 70, 71, 73, and 59, respectively. The number of unique OTUs in each level was 3, 1, 2, and 0, respectively, while there were 51 common OTUs for all the treatments, accounting for 18.68% of the total OTUs (Fig. 2).

## Diversity of AMF in sugarcane rhizosphere soil under different chemical fertilization rates

The impact of various fertilizations on the richness and diversity of AMF communities in the rhizosphere soils of different sugarcane varieties was observed. In the rhizosphere soil of sugarcane variety GT58, the Chao1 and Shannon indices were higher in T1, T2, and T3 compared to T4, and significant differences were found between T3 and T4 ($P = 0.036 < 0.05$). The diversity and richness exhibited a dynamic trend of increase,

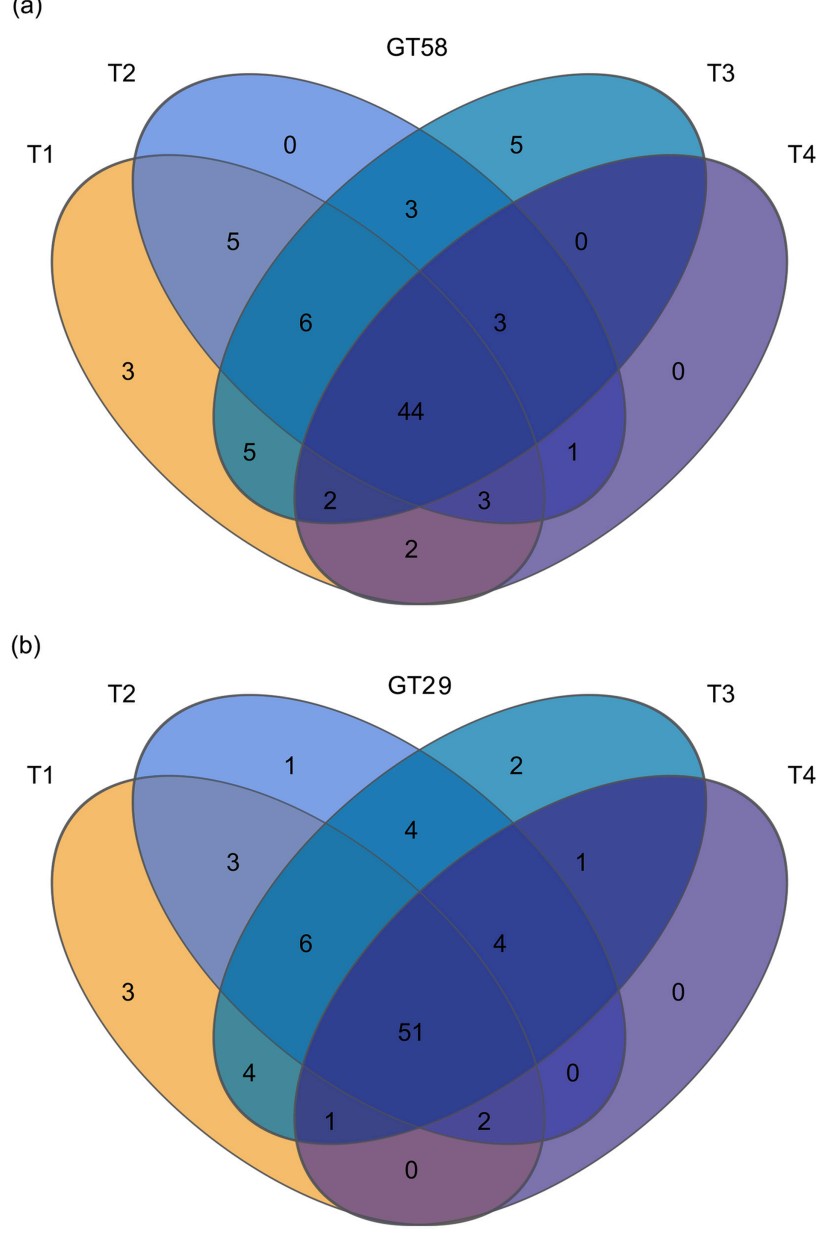

**Figure 2 Venn diagram of AMF communities in sugarcane rhizosphere soils under different chemical fertilization rates.** T1, T2, T3, and T4 represent the treatments with 0.0, 562.5, 1,125.0, and 2,250.0 kg/ha controlled-release compound chemical fertilizer with an N: P: K ratio of 17:7:17 (product of Hebei Tian-Ren Chemical Industry Co., Ltd., Qinhuangdao, China), respectively.

followed by decrease with increasing fertilization, reaching the highest at the T3 level. Conversely, the diversity and richness of AMF communities in the rhizosphere soil of sugarcane variety GT29 were not significantly influenced by different fertilization levels. This suggests that fertilization has a significant impact on the AMF community in the rhizosphere of GT58, but not in that of GT29 (Fig. 3).

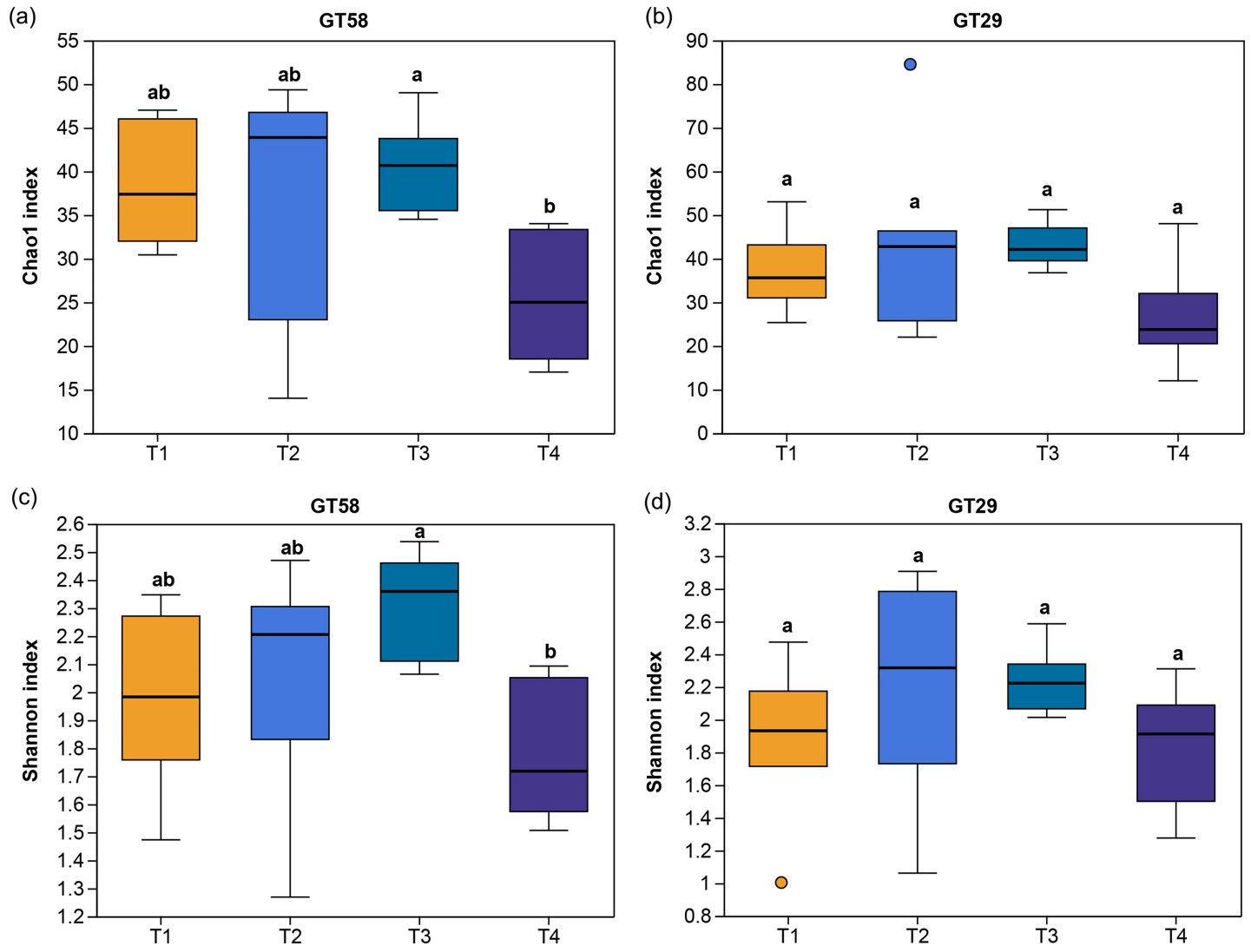

**Figure 3 Diversity of AMF in sugarcane rhizosphere soil under different chemical fertilization rates.** Different lowercase letters above the columns indicate the significant differences at $P < 0.05$. T1, T2, T3, and T4 represent the treatments with 0.0, 562.5, 1,125.0, and 2,250.0 kg/ha controlled-release compound chemical fertilizer with an N: P: K ratio of 17:7:17 (product of Hebei Tian-Ren Chemical Industry Co., Ltd., Qin-huangdao, China), respectively.                                

## AMF communities in sugarcane rhizosphere soil under different chemical fertilization rates

The composition of AMF communities in sugarcane rhizosphere soil under different chemical fertilization rates was illustrated in Fig. 4. For sugarcane variety GT58, the dominant genus of AMF in all the fertilization treatments was *Glomus*, with relative abundances exceeding 94%, and the highest abundance was observed in T3 fertilization treatment, reaching 99.74%. For different fertilization levels, T1, T2, T3, and T4 exhibited 4, 4, 5, and 3 fungal genera, respectively. The genera *Glomus*, *Acaulospora*, *Paraglomus*, and *Scutellospora* were present in T1, T2, and T3 treatments, while only *Glomus*, *Acaulospora*, and *Paraglomus* were found in T4 treatment. The genus *Diversispora* was

Peer J

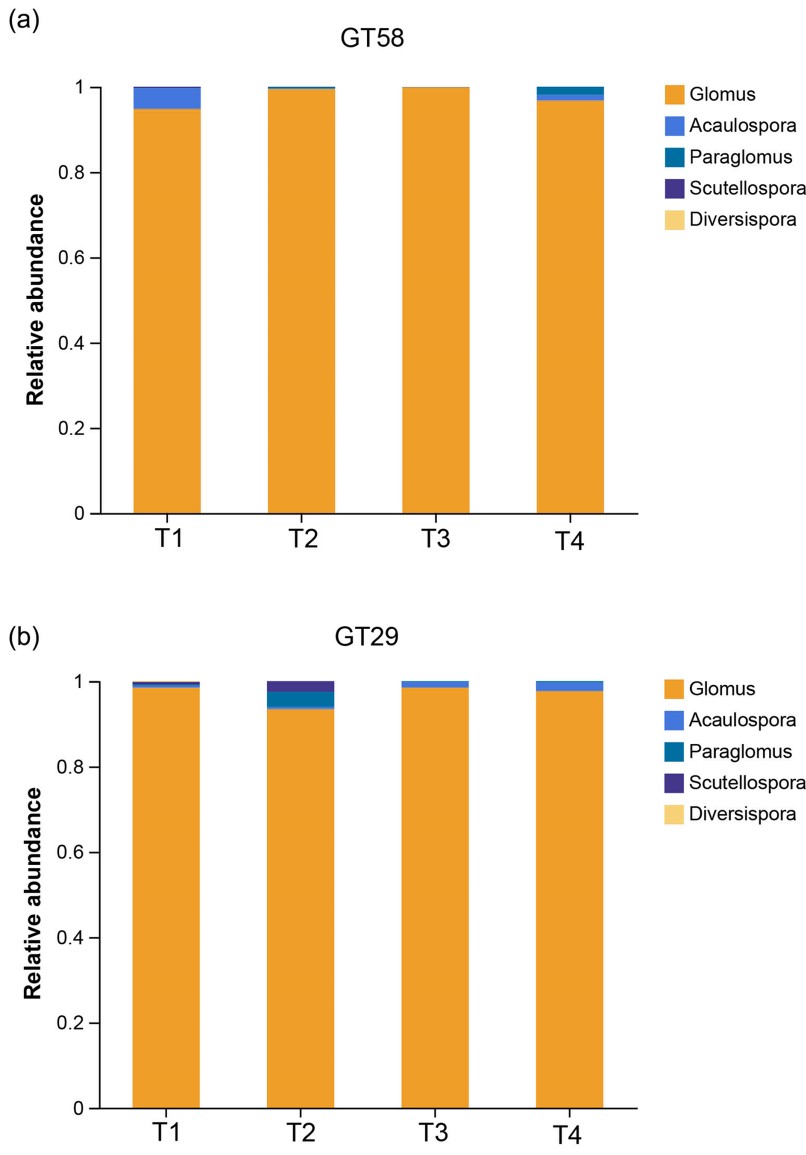

**Figure 4 The relative abundance of AMF communities in sugarcane rhizosphere soils with different fertilization rates.** T1, T2, T3, and T4 represent the treatments with 0.0, 562.5, 1,125.0, and 2,250.0 kg/ha controlled-release compound chemical fertilizer with an N: P: K ratio of 17:7:17 (product of Hebei Tian-Ren Chemical Industry Co., Ltd., Qinhuangdao, China), respectively.

exclusively present in T3 treatment. Similarly, for sugarcane variety GT29, the dominant genus of AMF was also *Glomus*, with relative abundances exceeding 93%, and the highest abundance was observed in T3 treatment, reaching 98.48%. For different fertilization levels, T1, T2, T3, and T4 exhibited 5, 4, 4, and 3 fungal genera, respectively. The genera *Glomus*, *Acaulospora*, and *Paraglomus* were present in all the fertilization treatments, while *Scutellospora* was present only in T1 and T2 treatments, with the relative abundances of 0.63% and 2.49%, respectively. The genus *Diversispora* was exclusively present in T1 and T3 treatments, with the relative abundances of 0.17% and 0.02%, respectively.

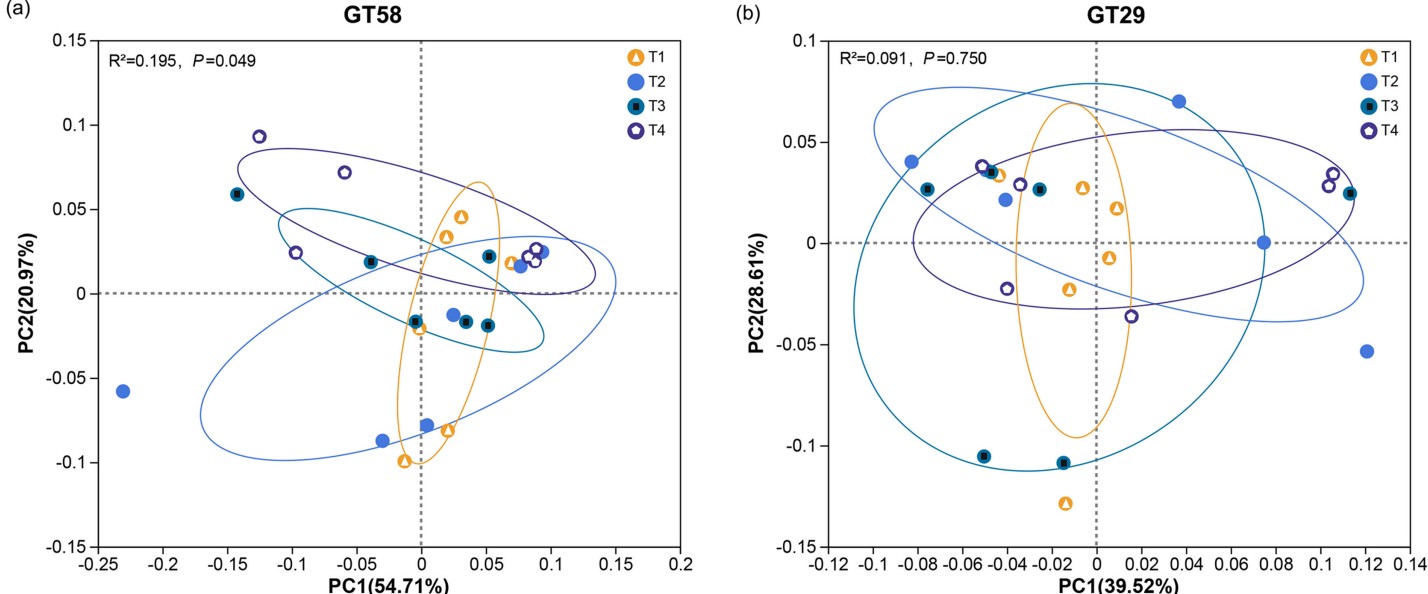

**Figure 5 PCoA analysis of AMF communities in sugarcane rhizosphere soil under different chemical fertilization rates.** T1, T2, T3, and T4 represent the treatments with 0.0, 562.5, 1,125.0, and 2,250.0 kg/ha controlled-release compound chemical fertilizer with an N: P: K ratio of 17:7:17 (product of Hebei Tian-Ren Chemical Industry Co., Ltd., Qinhuangdao, China), respectively.

## Discrepancy of AMF community in sugarcane rhizosphere soil under different chemical fertilization rates

Principal coordinates analysis (PCoA) was performed at OTU level to explore the similarity and dissimilarity in AMF community in sugarcane rhizosphere soils with different chemical fertilization treatments. As shown in Fig. 5, for sugarcane variety GT58, the PCoA 1 and 2 axes explained 54.71% and 20.97% of the variation, respectively. For sugarcane variety GT29, the PCoA 1 and 2 axes explained 39.52% and 28.61% of the variation, respectively. The samples of rhizosphere soil in GT58 with the same fertilization rate were clustered closely, indicating minor differences in the AMF community structure. Conversely, those in GT29 with the same fertilization level showed more dispersion, suggesting greater variability in the AMF community structure. The Adonis test results revealed a significant effect of chemical fertilization on the AMF community in the rhizosphere soil in GT58, but not in that in GT29 at $P = 0.753 > 0.05$ level.

## Distinctive OTUs in sugarcane rhizosphere soil under different chemical fertilization rates

Because the result that different chemical fertilization rates only significantly affected the AMF community with GT58, the subsequent analyses focused on identifying the distinctive OTUs in the rhizosphere soil of GT58 under different fertilization levels. The results revealed a total of 5 distinctive OTUs, all belonging to the genus *Glomus*. These OTUs were identified as OTU 41, OTU 74, OTU 85, OTU 122, and OTU 140. Among these distinctive species, the highest abundance was consistently observed in T3 treatment. Specifically, for OTUs 41 and 122, T3 exhibited significantly higher relative abundances

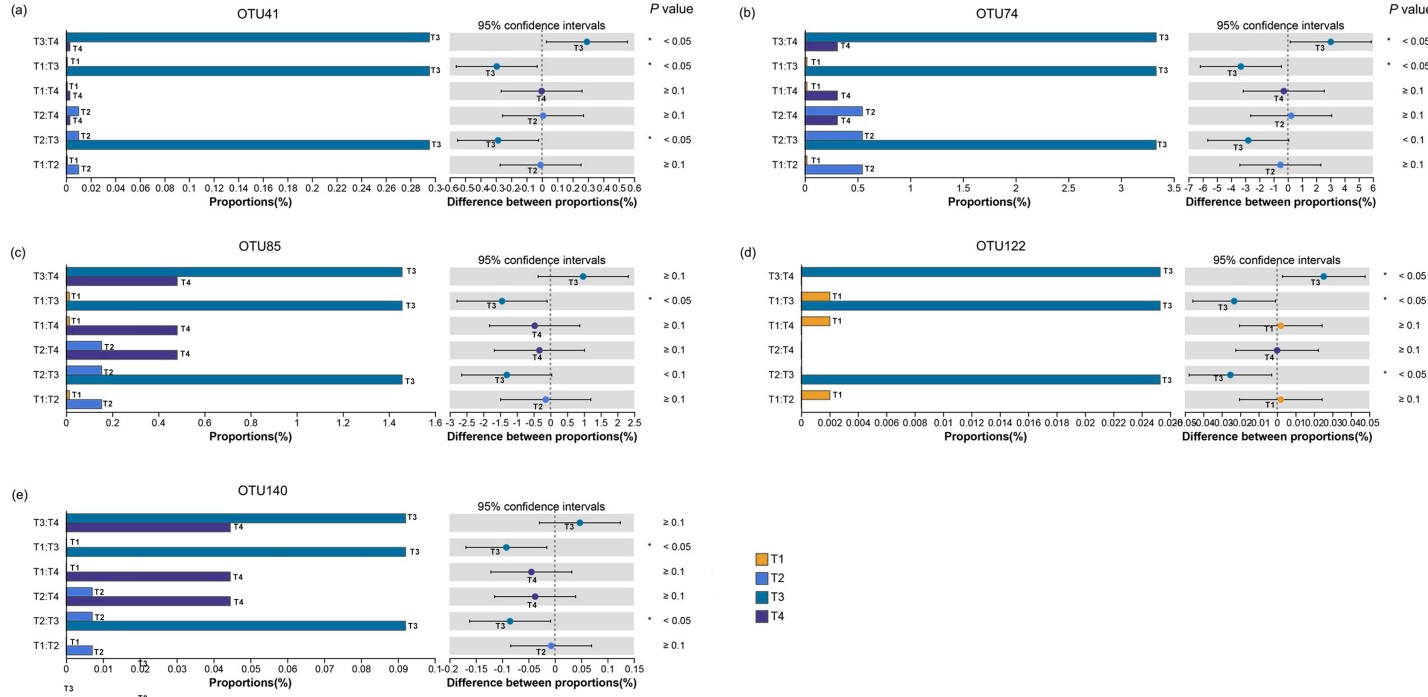

**Figure 6 Characteristic operational taxonomic units (OTUs) in sugarcane rhizosphere soil under different chemical fertilization rates.** T1, T2, T3, and T4 represent the treatments with 0.0, 562.5, 1,125.0, and 2,250.0 kg/ha controlled-release compound chemical fertilizer with an N: P: K ratio of 17:7:17 (product of Hebei Tian-Ren Chemical Industry Co., Ltd., Qinhuangdao, China), respectively.

compared to the other fertilization treatments, reaching 0.30% and 0.03%, respectively. For OTU 74, T3 (3.33%) showed significant differences from both T1 (0.02%) and T4 (0.31%). Significant differences were also detected between T3 and T1 for OTU 85, as well as between T3 and both T1 (0.00%) and T2 (0.01%) for OTU 140 (Fig. 6).

## Relationship between different AMF genera communities and agronomic traits

Pearson correlation analysis revealed significant positive correlations between cane yield and sprouting rate, plant height, and stalk diameter, while brix was significantly and negatively correlated with tiller number and millable stalks. Through Mantel tests between sugarcane agronomic traits and AMF genera community, it was found that the genus *Acaulospora* showed a highly significant positive correlation ($P = 0.000 < 0.001$) with millable stalks and a significant positive correlation ($P = 0.026 < 0.05$) with tiller number. The genus *Scutellospora* exhibited a significant positive correlation with sprouting rate. However, there were no significant correlations between the other AMF genera and sugarcane agronomic traits (Fig. 7).

## DISCUSSION

Cane yield is a comprehensive reflection of various agronomic traits and a key indicator of the quality of the population (*Franco et al., 2015*). The three major mineral nutrients (nitrogen, phosphorus, and potassium) are particularly crucial for sugarcane growth, yield,

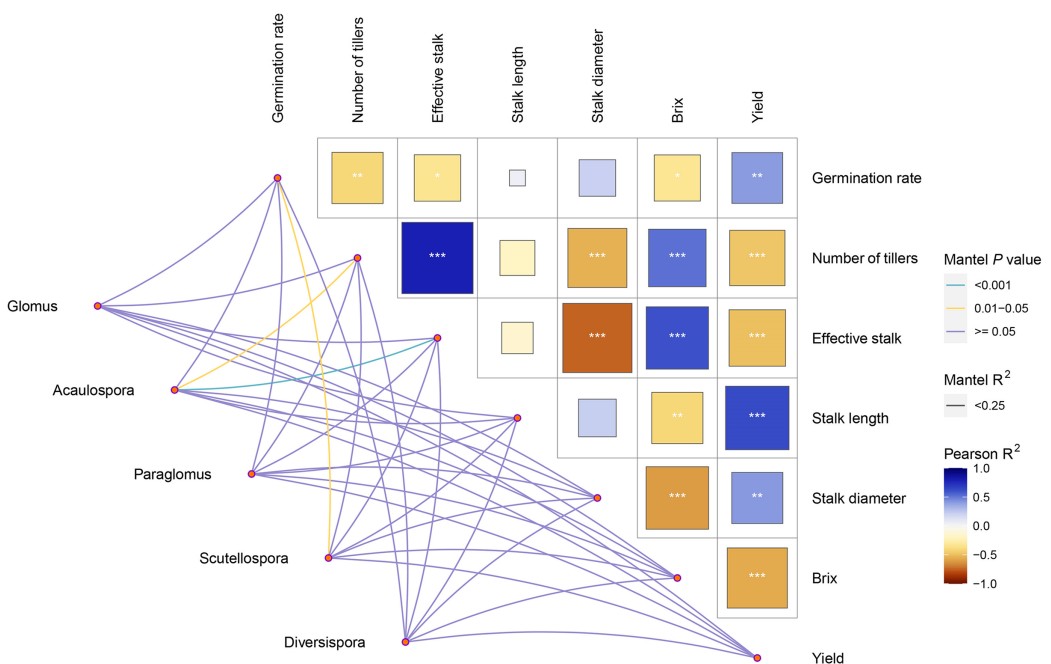

**Figure 7 The relationship among AMF genera community and agronomic traits.**

and quality formation (*Fortes et al., 2013*). Diverse levels of nitrogen, phosphorus, and potassium supply also affect the sugar content in cane (*Anderson, 1990*). Chemical fertilization, especially nitrogen application, is a method to enhance crop yield (*Singh et al., 2017*). Studies on nitrogen fertilization in sugarcane have shown that plant height and stalks diameter increase with increasing nitrogen fertilizer in a certain scope. However, excessive nitrogen application could not make the maximize yield, instead, it can lead to a decline (*Zhou & Ling, 2015*; *Brackin et al., 2015*; *Paungfoo et al., 2015*; *Vieria et al., 2015*). This aligns with the results of our study. Our findings indicate that chemical fertilization treatments have a positive effect on cane yield in sugarcane variety GT58 with treatments T2, T3, and T4, resulting in significant cane yield increases by 7.04%, 12.67%, and 11.11%, respectively, compared to the non-fertilized T1 treatment. Notably, there was no significant difference in cane yield between T2 and T3 or T4, suggesting that fertilization enhances cane yield, but the increase does not linearly correspond to the amount of chemical fertilizer applied. This could be attributed to nutrient imbalance caused by excessive nitrogen usage, resulting in reduced photosynthetic intensity and decreased resistance in sugarcane (*Xie et al., 2013*). However, in our study, different fertilization treatments had no significant impact on the cane yield in sugarcane variety GT29. This may be due to the variations in nitrogen uptake and use characteristics in different sugarcane genotypes (*Achieng et al., 2013*). As nitrogen application increases, the nitrogen use efficiency of ROC22 and YT00-236 significantly decreases, while the trend in fertilizer use efficiency for LC05136 differs from the other two sugarcane varieties, showing an increasing trend (*Li et al., 2019*).

Soil AMF communities are influenced by various soil environmental factors. Land use practices, soil properties and geographic distances impact the structure of AMF communities (*Hazard et al., 2013*; *Jansa et al., 2014*). Soil nutrient content can directly or indirectly affect AMF colonization. The results of this study show that different chemical fertilization treatments did not significantly affect the colonization rate but did have a significant impact on the colonization intensity and arbuscular richness. Related previous studies also demonstrated that chemical fertilization can significantly influence soil AMF diversity, and long-term nitrogen fertilization leads to a reduction in AMF diversity (*Jiang, 2019*; *Bhadalung et al., 2005*). It was reported that available phosphorus has the greatest impact on the abundance and diversity indices of AM fungi, and alkaline nitrogen and alkaline phosphatase are negatively correlated with the types of AMF (*Hu et al., 2016*). The results of this study revealed a dynamic pattern in the relationship between the diversity and richness of AMF in the rhizosphere soil of sugarcane variety GT58 and fertilization. As chemical fertilizer increased, the diversity and richness first increased and then decreased, with the highest in T3 treatment. This confirms that different fertilization treatments indeed affect AMF diversity. However, there have been studies suggesting that soil AMF diversity is not correlated with soil nutrient (*Guo et al., 2018*). *Hu, Luo & Wu (2015)* proposed that such discrepancies might be attributed to the differences in host plants, resulting in varying sensitivities of AMF to environmental factors. The results in this study also support this perspective as different fertilization levels had no significant impact on the AMF diversity and richness in the rhizosphere of sugarcane variety GT29.

Different chemical fertilization levels can indeed lead to changes in soil AMF communities. In this study, the detection of *Glomus* as the dominant genus aligns with other reports that also identified *Glomus* as a dominant genus within the Glomeraceae family (*Dai et al., 2013*; *Liu et al., 2017*; *Nong et al., 2023*). *Glomus* AMF dominate agricultural soils due to their physiological characteristics, allowing them to easily survive and reproduce through spores, mycorrhizal fragments, or mycelium (*Daniell et al., 2001*). Different AMF taxa respond differently to various fertilization levels, some significantly increasing and others decreasing. This suggests varying sensitivities of different fungal taxa to fertilizer inputs (*Bhadalung et al., 2005*). For example, the relative abundance of *Glomus* increases with the application of organic fertilizer (*Qin et al., 2015*). The Adonis test results in this study demonstrated a significant impact of chemical fertilization levels on the AMF community in the rhizosphere soil of sugarcane variety GT58. Relative abundance results also reveal that *Glomus*, a dominant genus in the AMF community, exhibited an increase in relative abundance with increasing chemical fertilization level, peaking in T3 treatment. A similar pattern was observed in sugarcane variety GT29. This phenomenon might be attributed to the fact that fertilization significantly increases the nutrient availability for microorganisms, thereby promoting the growth and reproduction of AMF that are preferentially favored. This can lead to changes in AMF community composition (*Oehl et al., 2010*).

The Mantel analysis results indicated a strong positive correlation between the genus *Acaulospora* and millable stalks, as well as a significant positive correlation between *Acaulospora* and tiller number. Similarly, a significant positive correlation was observed

between the genus *Scutellospora* and sprouting rate. These findings suggest that *Acaulospora* and *Scutellospora* can influence sugarcane agronomic traits, rather than the dominant genus *Glomus*, so plant growth is not solely determined by the relative abundance of AMF in the soil. Related research has found that the promotion of plant growth by AMF is often more effective when the root colonization rates are lower compared to the high colonization rates. This indicates that root colonization rate is not the sole indicator of their positive impact on plant growth (*Liu, 2016*; *Enkhtuya, Rydlová & Vosátka, 2000*). The primary factors influencing plant growth are related to the characteristics of the AMF themselves. One factor is the varying absorption capabilities of different AMF communities in terms of soil nutrients and water. Additionally, AMF exhibit selectivity towards host plants. These complexities warrant further investigation in subsequent studies.

## CONCLUSION

Different fertilization levels could significantly affect the cane yield in sugarcane variety GT58 and AMF community in the rhizosphere soil but had no significant effect on the diversity of AM fungal community in the rhizosphere soil in sugarcane variety GT29. Under T3 (1125.0 kg/ha) fertilization, the cane yield in GT58 was higher and the community of AMF in rhizosphere was more abundant as compared with the other fertilization treatments. It is recommended the T3 fertilization in sugarcane variety GT58 could be applied in commercial sugarcane production, which would be good to improve both the productivity of sugarcane and the community of AMF in the rhizosphere. This study also provides a theoretical basis for improving the stress resistance of sugarcane and the utilization rate of soil nutrients, reducing the fertilization rate, reducing the production cost of sugarcane by using AMF, and it would be helpful for the sustainable development of sugarcane industry.

### Funding
This work was supported by the Guangxi Natural Science Fund key projects (2019GXNSFDA245013), the Guangxi Academy of Agricultural Sciences Project (2021YT097; 2023YM96, 2024YP078, 2024YP016), and the Project of Guangxi Sugarcane Innovation Team of National Modern Agriculture Industry Technology System (gjnytxgxcxtd-2021-03-01), and Guangxi Key R&D Program Project (GK-AA22117009). Dr. Rong-Zhong Yang of Sugarcane Research Institute, Guangxi Academy of Agricultural Sciences provided guidance on the data analyses. This was their only role. The other funders had no role in study design, data collection and analysis, decision to publish, or preparation of the manuscript.

### Grant Disclosures
The following grant information was disclosed by the authors:
Guangxi Natural Science Fund key Projects: 2019GXNSFDA245013.

Guangxi Academy of Agricultural Sciences: 2021YT097, 2023YM96, 2024YP078 and 2024YP016.

National Modern Agriculture Industry Technology System: gjnytxgxcxtd-2021-03-01.

Guangxi Key R & D Program Project: GK-AA22117009.

## Competing Interests

The authors declare that they have no competing interests.

## Author Contributions

- Yi-Hao Kang conceived and designed the experiments, performed the experiments, analyzed the data, prepared figures and/or tables, authored or reviewed drafts of the article, and approved the final draft.
- Shang-Tao Jiang conceived and designed the experiments, performed the experiments, analyzed the data, prepared figures and/or tables, authored or reviewed drafts of the article, and approved the final draft.
- Qian Wang conceived and designed the experiments, performed the experiments, analyzed the data, prepared figures and/or tables, authored or reviewed drafts of the article, and approved the final draft.
- Ying-Jie Nong conceived and designed the experiments, performed the experiments, analyzed the data, prepared figures and/or tables, authored or reviewed drafts of the article, and approved the final draft.
- Juan Song performed the experiments, prepared figures and/or tables, and approved the final draft.
- Dong-Ping Li performed the experiments, prepared figures and/or tables, and approved the final draft.
- Yun-Ying Wen performed the experiments, prepared figures and/or tables, and approved the final draft.
- Jie Xu performed the experiments, prepared figures and/or tables, and approved the final draft.
- Ting-Su Chen conceived and designed the experiments, prepared figures and/or tables, authored or reviewed drafts of the article, and approved the final draft.
- Jin-Lian Zhang conceived and designed the experiments, prepared figures and/or tables, authored or reviewed drafts of the article, and approved the final draft.
- Yang-Rui Li conceived and designed the experiments, prepared figures and/or tables, authored or reviewed drafts of the article, and approved the final draft.

## Data Availability

The raw sequence reads are available at NCBI Bioproject: PRJNA103702. http://www.ncbi.nlm.nih.gov/bioproject/1037023.

## Supplemental Information

Supplemental information for this article can be found online at http://dx.doi.org/10.7717/peerj.17610#supplemental-information.

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
