# Peer review of "Response of soil arbuscular mycorrhizal fungal community to chemical fertilization in sugarcane"

_PeerJ, doi:10.7717/peerj.17610_

## Round 0.1 · original submission · Major Revisions

The authors are requested to address the issues raised by the reviewers and submit the revised manuscript for further consideration.

**Language Note:** PeerJ staff have identified that the English language needs to be improved. When you prepare your next revision, please either (i) have a colleague who is proficient in English and familiar with the subject matter review your manuscript, or (ii) contact a professional editing service to review your manuscript. PeerJ can provide language editing services - you can contact us at [email protected] for pricing (be sure to provide your manuscript number and title). – PeerJ Staff

·

Basic reporting

Manuscript is of good quality, based on the current need of research required, and having all the professional standard of writing and presentation facts.
Few minor improvement is needed in the manuscript as per following:
Line 43: replace 'resistance' with 'tolerance'
Line73: replace 'approximately' with 'about'
Line 73-74: add reference supporting the statement
Line 145: elaborate the protocol & procedure for estimation with supporting reference if available
Line 149: elaborate five-point sampling methods, add supporting reference
Line 360: Remove underline - Acaulospora

Experimental design

Experiments were conducted in a well defined and planned way. research gaps has been addressed in nice way.

Validity of the findings

Research gaps has been addressed in nice way. Findings of the this will have potential to be applied in the sugarcane cultivation for judicious use of fertilization and also increased plant vigor, growth and yield.

Additional comments

Besides studies on agronomic trait, studies on physiological traits could be included. However, the findings of the investigation addressed all the research gaps in this study.

Reviewer 2 ·

Basic reporting

Ref: Peer J 92939
Title: Response of soil arbuscular mycorrhizal fungal community to chemical fertilization in sugarcane
Please carefully read it once more, revise the introduction section. Cite recent papers as well. Although, authors framed the manuscript well but need to improve the description in materials and methods section to provide more justification of the study.

The diversity and richness of AMF communities in the rhizosphere soil of sugarcane variety GT29 were not significantly influenced by different fertilization levels while it was influenced in variety GT58, any specific reason, whether GT29 provide favourable environments in rhizosphere to AMF communities or other factors may involve ????.

The genera Glomus, Acaulospora, Paraglomus, and Scutellospora were present in T1, T2, and T3 treatments, while only Glomus, Acaulospora, and Paraglomus were found in T4 treatment. It shows that AMF communities increased as per fertilizer dose. Similarly, genus Diversispora was exclusively present in T3 treatment. Author should be clarified the reason behind this.

The samples of rhizosphere soil of GT58 with the same fertilization rate were clustered closely, indicating relatively small differences in AMF community structure. Conversely, those of GT29 in the same fertilization level showed more dispersion, suggesting greater variability in AMF structure. Whether author find any variations in phytochemical traits present in GT58 and GT29 variety. It may be possibility that crop may release such type nutrients which favor the growth of AMF.

Conclusion: no conclusion section in manuscript?????

General
The author should check all citation and references in entire manuscript and arrange as per the format of the journal. All the scientific name should be in italics form, name of the journal citation, figure, tables should be uniform.

Decision: I must be recommended for major corrections for further revision and acceptance.

Experimental design

satisfied

Validity of the findings

satisfied

·

Basic reporting

'no comment

Experimental design

no comment

Validity of the findings

no comment

Additional comments

The article titled "Response of soil arbuscular mycorrhizal fungal community to chemical fertilization in Sugarcane" authored by Yi-Hao Kang et al. provides valuable insights into the influence of chemical fertilization on the arbuscular mycorrhizal fungi (AMF) community in sugarcane rhizosphere soil. The authors investigate the effects of different chemical fertilization rates on cane yield and the diversity of root-associated AMF in sugarcane varieties GT58 and GT29.
Overall, the study underscores the significant influence of chemical fertilization on cane yield and the AMF community in sugarcane rhizosphere soil, emphasizing the importance of considering different sugarcane varieties and AMF diversity when implementing fertilization strategies. However, minor revisions are required to enhance clarity and adherence to reporting standards.

To achieve these improvements, I recommend the following

For instance, the abstract refers to both "cane yield" and "sugarcane yield." Choose one term and stick with it for clarity and consistency.
97; Provide a clearer justification for why the study is necessary. Why is it important to investigate the effects of different chemical fertilization methods on sugarcane growth and AMF communities? How will the findings contribute to addressing the problems of excessive chemical fertilizer application and improving sugarcane production?

107: While it mentions the location (Li-Jian Scientific Research Station), it lacks information on specific environmental conditions during the experiment (temperature, humidity, etc.). The section briefly mentions the planting date and number of buds planted for each variety but lacks information on planting depth, spacing between individual plants, and other relevant details that could influence plant growth.

147; While it mentions the depth of soil collection (0-30 cm), it doesn't provide a rationale or justification for choosing this depth. It doesn't mention how many replicate samples were collected at each sampling point.

140; Ensure that terms such as "millable canes," "stalk diameter," and "brix" are defined or briefly explained. Please consult the paper for this section: https://www.mdpi.com/2076-2607/9/10/2008

177; Provided primer sequences for AMF DNA amplification but lacks information on the region targeted, lacks details on the specific PCR amplification conditions for AMF DNA.

192; While the statistical tests are mentioned (e.g., Adonis test, one-way ANOVA, Mantel tests), there could be more explanation of why these tests were chosen and how they contribute to the overall analysis.

Line 211-212: The significance levels (P > 0.05, P < 0.05) are mentioned, but it would be more informative to provide actual p-values for each comparison.

Line 212: The statement "No significant differences were observed between T4, T3, and T2, however, both T4 and T3 showed significant increases than T1 by 11% and 10%, respectively" provides clear information.

Line 224: The statement "For GT29, T3 exhibited the highest brix, which was 6% higher than that in T2" provides a clear comparison. It might be helpful to include the actual values of brix for T3 and T2.

295: What is AMF?

In the discussion section, Functional Soil Microbes-found in your study discuss sugarcane important microbial taxa affected by monoculture that could provide additional context for understanding the intricate interactions between soil microbes, fertilization practices, and sugarcane productivity. However, these microbes under different sugarcane cultivars and cropping times have been found in recent studies of Chinese sugarcane cultivars (Regan14-62, Guitang 08-120, Haizhe 22, Guitang 08-1180, Taitang 22 and Liucheng 05-136). These functions can be found here:

- https://link.springer.com/article/10.1007/s13205-021-03091-1
- https://www.ncbi.nlm.nih.gov/pmc/articles/PMC8724486/

---

## Round 0.2 · Minor Revisions

The manuscript is revised and authors have done wonderful efforts. However, our editorial team is of the opinion that the contents of the paper are congruent. But could the authors please add the implications and the scientific and technological contributions to the conclusions?.

·

Basic reporting

The revised manuscript has addressed all comments and is now ready for publication.

Experimental design

no comment

Validity of the findings

no comment

Additional comments

no comment

---

## Round 0.3 · accepted · Accept

The manuscript is accepted in its current form.